# Advances in Targeted Delivery of Doxorubicin for Cancer Chemotherapy

**DOI:** 10.3390/bioengineering12040430

**Published:** 2025-04-19

**Authors:** Wenhui Xia, Martin W. King

**Affiliations:** Wilson College of Textiles, North Carolina State University, Raleigh, NC 27606, USA; wxia5@ncsu.edu

**Keywords:** drug delivery, cancer, nanotechnology, carrier, targeted therapy

## Abstract

Doxorubicin (DOX) is one of the most powerful chemotherapy drugs used to treat different kinds of cancer. However, its usage has been limited by typical side effects and drug resistance, particularly cardiotoxicity. According to studies, a more effective and promising method is to conjugate it or entrap it in biocompatible nanoparticles. Compared to free DOX and traditional formulations, nanoparticles using specific processes or techniques can improve drug stability, minimize premature release at untargeted locations, and lower systemic toxicity. This review explains how various nanocarriers target the tumor to improve therapeutic efficacy while reducing the negative effects of DOX.

## 1. Introduction

The second most common cause of mortality globally, after cardiovascular illness, is cancer, which is characterized by the aberrant and uncontrolled development of tumor cells. Currently, invasive methods of treating cancer include radiation therapy, surgery to remove the tumor if possible, and primary chemotherapy to decrease existing cancerous tumors. However, every technique has limitations [1,2]. Chemotherapeutic drug resistance and adverse side effects from buildup in healthy tissues limit the use of chemotherapy. More specifically, chemotherapy also sacrifices the ability of normal cells, like bone marrow and hair follicles, to grow, proliferate, and maintain their viability [3].

A common and well-researched anticancer medication, doxorubicin (DOX) is an essential part of many chemotherapy treatments. It has been one of the most widely used anticancer agents. The chemical structure of this anthracycline consists of four anthraquinone rings joined to a single amino sugar moiety. DOX performs several functions, including suppressing topoisomerase II, inhibiting DNA unwinding, and facilitating intercalation with DNA. Its ability to obstruct transcription and DNA replication is thought to be the source of its anticancer properties [4]. Unfortunately, standard chemotherapy with DOX also has many drawbacks. DOX delivery may cause cellular damage in healthy cells as well, which leads to myelosuppression, nausea, and vomiting shortly after administration [5]. Additionally, research has demonstrated that the addition of DOX to traditional chemotherapy might result in cardiotoxicity by raising oxidative stress, which damages heart tissue and causes cardiomyopathy [5].

Because of developments in nanotechnology that allow for the creation of nanoparticles with varying sizes, shapes, surface characteristics, and in vitro/in vivo behaviors, targeted drug delivery systems have revolutionized cancer treatment in the recent decades [6]. Numerous cytotoxic drug carriers, including peptides, antibody-drug conjugates, and nanocarriers, are being studied for use in both active and passive cancer treatment. They have many beneficial properties, including improving the permeability and retention of chemotherapy drugs, functionalizing the carrier surface, and above all, properly transporting the drugs to the intended location [3]. It is also asserted that creating chemotherapy drug delivery systems based on nanotechnology can improve drug solubility and bioavailability, increase chemotherapeutic agents’ biodistribution, avoid multidrug resistance, and reduce non-specific toxicity [3]. Therefore, reducing the possibility of adverse effects may be possible by developing a multifunctional nanocarrier for targeted drug delivery that can transport and deliver DOX to cancer locations [7].

This study outlines advances in DOX-loaded nanocarriers for efficient and targeted cancer treatment, as well as an explanation of the mechanics and research examples for each approach. Furthermore, the disadvantages of present available products and methods are explored, and future development possibilities are given.

## 2. Doxorubicin (DOX)

For many types of cancer, DOX is a powerful chemotherapy medication. It is most frequently used to treat solid organ tumors such as breast and stomach cancer as well as hematological cancers such as lymphoma. DOX is a part of the well-known anthracycline antibiotic family, which also includes daunorubicin, epirubicin, idarubicin, and others [8]. Following a structural alteration of daunorubicin, it was identified in 1969 and showed less lethal cardiac toxicity than daunorubicin [9]. The tetracene quinone ring structure of anthracyclines, which belong to the polyketide family, is joined to a sugar molecule by a glycosidic bond. As illustrated in Figure 1, DOX is amphiphilic, having a hydrophilic amino-sugar functional group daunosamine (C_6_H_13_NO_3_) and a hydrophobic aglycone group adriamycinone (C_21_H_18_O_9_) [8].

### 2.1. Mechanism of Action

Numerous investigations have demonstrated that the DOX capacity to intercalate into the DNA duplex is the primary mechanism by which it effectively fights cancer [10]. Intravenous injection is the most well-known method of administering DOX. DOX has a distribution half-life of only three to five minutes and is quickly absorbed by cells after injection. Plasma proteins can be bound by DOX and its product, doxorubicinol, and then transported to tissues. Additionally, the nucleus of cells is a more likely location for DOX than the cytoplasm [11].

### 2.2. Limitations

#### 2.2.1. Side Effects

Fast elimination from the body and a low tumor-reaching rate are two drawbacks of doxorubicin therapy that necessitate greater dosages. Therefore, DOX has a deadly effect on immunological and other normal cells in addition to cancer cells, resulting in serious clinical drawbacks and undesirable side effects, such as extreme toxicity and inflammation in healthy organs [12]. As a result, the drug has a significant level of heart toxicity. Following cardiac toxicity, doxorubicin cardiomyopathy develops, which often leads to heart failure. The DOX peak plasma concentration determines this toxicity. At a total dose of 400 mg/m^2^, Von Hoff et al. observed a 3% incidence of free DOX-induced congestive heart failure; this might increase to 7% at 550 mg/m^2^ and 18% at 700 mg/m^2^ [8]. Alopecia, nausea, bone marrow suppression, and persistent exhaustion are some of the other severe adverse effects of DOX [9].

#### 2.2.2. Drug Resistance

It is given either as a single-use infusion or as a follow-up intravenous dosage after 21 days. The suggested dosage is between 50 and 75 mg/m^2^, and 500 mg/m^2^ is regarded as a safe cumulative dose [8]. However, one major issue is DOX resistance, which either results in higher dosages during treatment or the usage of multiple concurrent medications which increases the drug’s toxicity [1]. Therefore, the strategy of loading DOX on a nanocarrier is an attractive approach as it simultaneously reduces toxicity and increases therapeutic efficacy.

## 3. Strategies Used in Cancer Therapy

### 3.1. Loading Methods

Nanoparticles (NPs) range in size from 1 to 100 nanometers [9]. Covalent bonds or physical adsorption, such as electrostatic contacts between the medication and the nanoparticle, can bind drugs used to treat cancer to the outer shell or the corona of NPs (Figure 2). These connections typically show poor stability and become pH-sensitive. Non-covalent binding is easy to apply and adaptable, and it causes little alteration to the medications’ structure or biological function. In addition, the medication reacts to environmental stimuli with ease and speed. The regulated release of the pharmaceuticals is dependent on the linkers used, and the amplified effective concentration of the drugs is much increased at the target site when the drug forms a covalent bond with the nanocarriers, resulting in improved solubilization [13].

Hydrogen bonds, electrostatic contacts, or hydrophobic interactions may also trap medications within the core of NPs, preventing them from being hydrolyzed or digested by enzymes and slowing down the immune system’s immediate response and clearance [13,14]. Additionally, it can reduce the rate of medication administration and considerably extend the drug’s half-life in vivo [14].

### 3.2. Targeting

When nanocarriers are the right size, they can use enhanced penetration retention (EPR) of the tumor tissue to passively target and distribute medications (Figure 3). Drug trapping, drug accumulation, and preferential uptake by the leaky vasculature in the cancerous region are caused by the unique properties of tumor tissue, specifically uneven endothelial cell gap formation and poor lymphatic drainage [1].

However, total reliance on passive targeting has serious drawbacks, including the potential for NPs to accumulate in the liver and spleen due to their fenestrated vasculature and the incapacity of the NPs to adequately penetrate deeply enough through the intricate tumoral network because of structural heterogeneities. Therefore, it became essential to create systems that use both passive and active targeting techniques. Drug distribution that is reliant on molecular identification to a specific spot is necessary for active targeting. Given that tumor cells overexpress receptors involved in growth and survival pathways, nanocarriers can be coupled to natural ligands, like monoclonal antibodies, folate, and peptides, to target these receptors, causing them to be accumulated in the localized cancer cell tissue environment [1,15]. The type of ligand-receptor interaction can affect the rate of cellular internalization. The addition of a targeting moiety frequently facilitates the drug’s cellular absorption through receptor-mediated endocytosis. A number of research teams are working to improve NPs’ targeting capabilities and to develop targeted treatments based on these principles [1].

## 4. Nanocarriers Used for Targeted DOX Delivery

The two types of nanocarriers are classified as organic and inorganic depending on their composition. The majority of organic nanocarriers, such as micelles, liposomes, and polymeric nanoparticles, are biodegradable. They are recognized for having both biocompatibility and a high drug-loading capability [3]. Metals, like gold nanoparticles, iron oxide, magnetic iron, and nickel and cobalt compounds, are typically used to create inorganic NPs [9]. Because of their many advantages, including a significant drug loading capacity, a sizable surface area, decreased in vivo toxicity, and increased bioavailability, inorganic nanocarriers are studied extensively as drug delivery agents. However, they are also associated with some disadvantages, including a lengthy half-life in vivo, slow-release rates, and non-biodegradability (Table 1) [3].

### 4.1. Liposomes

Because liposomes are spherical nanoparticles that range in size from 30 to 500 nm and have both a lipid bilayer and an aqueous compartment, they can be used to transport both hydrophilic and hydrophobic drug candidates, making them the perfect candidate for a variety of medications with different solubility levels [3,6]. This is because natural phospholipids make up the bilayer structure of liposomes. The phospholipids’ hydrophobic tails point inside, while their hydrophilic heads point outwards (Figure 4). This unique structure allows for the incorporation of hydrophilic molecules in the core and enables the external lipid membrane to contain hydrophobic medications [13].

Because liposomes resemble cell membranes structurally, they are highly biocompatible and biodegradable. To maximize the dosage for medication delivery and improve its therapeutic impact, DOX can be physically encapsulated inside liposomes. This approach not only shields DOX from enzymatic destruction before it reaches the lesion site, but it also increases its stability and lowers its toxicity. In addition, ligands or other functional groups can be added to the phospholipid bilayer’s surface either chemically or physically so as to make the liposomes more tissue targeted. This will increase the liposome’s effective retention period at the lesion site and enhance its delivery efficiency [14]. Patients treated with liposomal DOX show less myelosuppression, cardiotoxicity, nausea, and vomiting than those treated with traditional DOX systemically [16].

Biocompatible and inert polymers like polyethylene glycol (PEG), have been added to the surface of liposomes to provide a protective barrier that delays the normal reticuloendothelial system (RES) macrophages from removing drugs and pathogens from the body. This lengthens the period of time when the DOX drug is available and can function as a chemotherapeutic agent [13]. Compared to ordinary liposomes, pegylated liposomes are not usually absorbed by RES cells which facilitate the removal of DOX from the circulation. Pegylated liposomal DOX has been found to exhibit superior therapeutic activity and higher intratumoral concentrations of the drug when compared to standard (non-pegylated) liposomes of DOX. However, non-pegylated liposomal DOX injections are safer than both pegylated liposomal DOX and conventional DOX [9].

To verify the safety and efficacy of the nanomaterials, the nanomedicine product must undergo preclinical testing on animals before undergoing Phase I, Phase II, and Phase III of human testing [17]. Only a small number of nanocarriers have been approved for clinical use, thus even though they have demonstrated remarkable promise in clinical trials, their successful transition into clinical practice is still difficult [3]. Currently, only pegylated (Doxil^®^, Lipodox^®^) and non-pegylated (Myocet^®^) versions of liposomal DOX have been studied and approved by the US Food and Drug Administration (FDA) for systemic delivery as chemotherapeutic DOX nanodrugs (Table 2). Prior phase III clinical trials of ThermoDox^®^ failed to demonstrate statistically significant improvement in the primary endpoint, despite the fact that it was one of the most promising ongoing clinical investigations [18]. This led to the temporary suspension of the ThermoDox^®^ study [19].

Both passive and active loading techniques have been typically employed for the encapsulation of lipophilic DOX within liposomes. Although passive loading, which usually involves a thin-film hydration method, is straightforward, it produces very low drug encapsulation and loading levels because the drug solution is added during the liposome formation process [15]. Conversely, active methods, also referred to as distant loading methods, use a pH gradient between the exterior buffer carrying the DOX solution under neutral physiological conditions and the interior acidic core buffer of the blank liposome [28]. In order to deliver the DOX, Myocet^®^ employs a citrate transmembrane gradient approach, which involves changing the pH of the intraliposomal fluid from a 300 mM citrate buffer at pH 4.0 to a 4-(2-hydroxyethyl)-1-piperazineethanesulfonic acid (HEPES) buffer at pH 7.4. This creates an electrical potential across the cell membrane and achieves an up to 95% encapsulation efficiency. This extends the delivery period for the DOX-citrate bundled fiber complex and prevents medication leakage and premature release [15].

Similarly, Chowdhury et al. prepared twelve liposomal batches by thin film hydration using various saturated and unsaturated lipids. They then loaded the liposomes using the ammonium sulfate method to improve the delivery of DOX to human epidermal growth factor receptor 2 positive (HER2-+) breast cancer cells. The entrapment efficiency of the optimal formulation of the DOX was around 93%. Research indicates that for the HER2-+, Michigan Cancer Foundation (MCF-7) and Sloan-Kettering breast cancer (SKBR-3) cells, the DOX-loaded liposomal formulations had a lower half-maximal inhibitory concentration (IC50) than the systemically intravenous injected DOX, and are more effective in killing cancer cells [29]. The IC50 value is routinely used to determine a drug’s potency. A lower IC50 implies a more effective cancer inhibitor, as it requires less medication to provide the same inhibitory effect.

### 4.2. Micelles

Micelles, which are aggregations of amphiphilic surfactant molecules, are another kind of lipid nanostructure with particle sizes ranging from 5 nm to 100 nm. They are divided into two groups: polymeric micelles and low molecular weight surfactant micelles. Because the amphiphilic molecules self-assemble in an aqueous solution, they spontaneously aggregate into spherical vesicles to create micelles, which have a non-polar hydrocarbon chain in the tail that can be embedded in the center. Amphiphiles self-assemble and aggregate into micellar nanoparticles, which are held together by van der Waals bonds. This only happens when the concentration of the surfactant surpasses the critical micelle-forming concentration (CMC) and the system temperature surpasses the critical micelle temperature (CMT) or Kraft temperature. Micelles can be formed from fatty acids, phospholipids, fatty acid salts (soaps), and other such structures. Depending on the solvent type, the blocking chain length, the temperature, and the type of blocking agent, micelles can have different shapes or morphologies, namely spherical, rod-like, tube-like, capsule-like, or flat like a sheet [1,4].

The hydrophilic head of polymeric micelles creates the exterior shell, while the hydrophobic tail forms the inside and shields hydrophobic medications from the environment. While the shell acts as a stabilizing barrier that improves spatial stability and permits versatile functionalization, targeted drug delivery is maintained through control of pH, temperature, ultrasound and surface modifications with ligands like peptides or folates. At the same time, the hydrophobic core can be used to encapsulate a large amount of a hydrophobic drug at higher concentrations. This means that the DOX can be loaded more rapidly and the drug solubility improved significantly [13,14]. The nanocarrier system can be stable and durable in the bloodstream, and the RES cells in the blood circulatory system find it difficult to identify and trap the micelles [14]. Another important feature is their biocompatibility, which permits degradation in the body without generating a toxic response [4].

An appropriate physiological stimulus for pH-responsive drug delivery can be provided by the fluctuations of pH in different parts of the human body, which enables it to distribute the active chemotherapeutic agent to the area of interest [5]. Because of decreased blood flow and increased aerobic glycolysis, the malignant tissue is mildly acidic. To release drugs that target tumors, pH-sensitive nanoparticles have been developed in recent years. They are stable enough at a physiological pH of 7.4 but release the active ingredient when the pH trigger point is reached [30]. This is accomplished by giving the nanostructures acid-sensitive linkages that allow them to break down in the acidic environment of tumors. Ionizable groups are integrated into the design of the nanoparticles in pH-dependent nanoscale delivery systems, which are based on protonation and ionization. Protonation, or charge reversal, takes place at low, acidic pH values, changing the hydrophobic and hydrophilic characteristics of the nanoparticles and causing the medication to be released. When creating micelles, ionizable groups such as amino, carboxyl, sulfonate, and imidazolyl are generated and allow the DOX drug to be released [4].

pH-sensitive micelles have been shown to enhance DOX’s anticancer activity both in vitro and in vivo. A type of DOX-conjugated pH-sensitive polymeric micelle containing poly (L-lysine) 4-carboxy benzaldehyde doxorubicin grafted block poly (methacryloyl-oxyethyl phosphoryl choline (PLL(CB/DOX)-b-PMPC) was created by Ma et al. in one investigation. In contrast to less than 50% release over the same period at pH 7.4, DOX release in acidic conditions (pH 5.5) approached 40% within 2 h and reached 80% after 48 h, indicating that drug activation occurred largely within the tumor cells. Micelle-treated tumor-bearing mice demonstrated a considerable decrease in tumor volume in vivo, with a tumor size reduction of around 50% compared to the systemically intravenous injected DOX and a markedly reduced systemic toxicity [31].

Furthermore, because folic acid (FA) selectively binds to the folate receptor (FR), which is overexpressed in cancer cells, FA has been described as an effective targeted prodrug [5]. FA was linked to the polyethylene glycol 5K Daltons-embelin2 (PEG5K-EB2) micelle surface by Lu et al. to enhance the targeted delivery of DOX to tumors. With sustained release kinetics, the DOX loading efficiency (DLE) for PEG5K-EB2 and FA-PEG5K-EB2 micelles was 91.7% and 93.5%, respectively. The potential of FA-PEG5K-EB2 micelles for targeted, effective and safer DOX delivery in cancer therapy was demonstrated by in vivo tests that showed a tumor inhibition rate (IR) of 85.45% for these micelles, which was higher than that of free DOX (44.22%) and Doxil, which is a form of doxorubicin HCl encapsulated in liposomes for intravenous use (66.97%) [32].

### 4.3. Polymeric Nanoparticles

Colloidal systems are polymeric nanoparticles. These are assemblages of organic polymer compounds that take the shape of either hollow spheres with void space in the middle, known as nano-capsules, or as solid spheres, known as nanospheres. They can consist of synthetic polymers like polyvinyl alcohol (PVA) and poly (lactic-co-glycolic acid) (PLGA), as well as natural polymers including chitosan and cellulose [13]. These polymeric core–shell nanostructures make it easier to encapsulate hydrophobic medications, prolong their half-lives, and regulate their release. It is also comparatively simple to manipulate the physicochemical characteristics of polymers, such as their molecular weight, flexibility, crystallinity, and surface charge, so as to produce effective biodegradability, superior drug loading at targeted tumor locations with lower cytotoxicity and increased efficacy. Because their surfaces can be stabilized by conjugating, grafting, cross-linking and adsorbing hydrophilic polymers, these systems have unique characteristics such as a longer circulation life and decreased hepatic absorption [13,33].

#### 4.3.1. Nanogels

Cross-linked nanoparticles called nanogels are made from hydrophilic polymers which expand when water is added and enable spontaneous drug loading in aqueous conditions. They may be created by combining natural or synthetic polymers, cross-linking them covalently or noncovalently using hydrogen bonds, and utilizing hydrophobic or electrostatic interactions [2]. As a polymer-based drug delivery system, nanogels provide the benefit of being able to be adjusted in size from nanometers to micrometers, which can be used to prevent phagocytic cells from identifying them and removing them [33]. Compared to liposomes or polymeric micelles, they also have a very high DOX-carrying capacity because of their intrinsic porosity [14]. In addition, unlike other drug delivery systems, nanogels have good biocompatibility and can be delivered by oral, nasal, pulmonary, intraocular, and parenteral routes [33].

For targeted DOX delivery, Xu et al. synthesized alginate and poly(N-isopropyl acrylamide) (PNIPAM) nanogels that are thermosensitive, pH-, and redox-responsive. The nanogels showed a high loading capacity of 6.8% and an encapsulation effectiveness of 91.7%. The DOX release increased under acidic conditions (pH 5.0), reaching 76% after 6 h, but it was slower at physiological pH (7.4), reaching only 64% over 72 h. Utilizing the thermo-sensitivity of these nanogels to rupture endosomal vesicles, in vitro cytotoxicity tests employing the oral adenosquamous carcinoma cell line (CAL-72) demonstrated increased cell death with temperature changes, attaining an IC50 of 0.35 µM as opposed to systemically intravenous injected free DOX’s IC50 of 2.10 µM [34].

#### 4.3.2. Dendrimer

Dendrimers are monodispersed polymers with a highly branching structure and a highly symmetric spherical form. They typically have a diameter of 1–10 nm [6,13,14]. They are often made from synthetic or natural materials, such as sugars, amino acids and nucleotides [13]. Typically, they have a core molecule which is covalently bonded to three-dimensional branched structures. The branches can be added to the central portion through synthesis or polymerization of the central molecule. The branches can form a small and compact sphere due to their limited size [1]. Low generation dendrimers have a smaller size, a flexible shape, a more open structure, less branching, and have fewer surface functional groups compared to high generation dendrimers, which are larger, more compact, have more branching in their globular structure, and a greater number of exposed functional groups on their surfaces [13].

The surface of high generation dendrimers can have DOX covalently bonded to them. When the dendrimers’ surface is heavily coated with different functional groups, they can bind to target cells, delivering DOX to the right place and facilitating targeted chemo-therapeutic action [13,14]. In addition, DOX may be held in the dendrimer core by hydrophobic or electrostatic interactions, as well as hydrogen bonds. It is possible to regulate the solubility and chemical activity of these macromolecules by affixing particular functional groups to their surfaces [1,13].

For a variety of material science and biotechnology applications, polyamidoamines (PAMAM) are a family of commonly utilized dendrimers composed of repeatedly branched amide and amine groups. They come in a series of generations (G 0–10), with five different cores and 10 external surface classes [1]. PAMAM’s distinct molecular structure and consistent size range make it an appropriate drug delivery vehicle [6].

The specific receptor of alpha-fetoprotein (rAFP3D) is expressed on the surface of many tumor cells but not in healthy human tissues. Yabbarov et al. added a polymeric carrier to the second generation (PAMAM G2) dendrimer. By conjugating an acid-labile linker to the dendrimer, the DOX loading was significantly improved. At physiological pH (7.4), the DOX release from the conjugated rAFP3D-G2-DOX was comparatively slow, releasing only 8% over 24 h. However, at lower pH levels, the release was accelerated. With half-life values ranging from 121 min at pH 5 to 687 min at pH 6, 90% of the DOX was released at pH 5.5 in 24 h (Figure 5). In comparison to systemically intravenous injected free DOX, this conjugated rAFP3D-G2-DOX controlled release compound achieved five times greater absorption by facilitating selective accumulation in the human ovarian cancer cell line with epithelial-like morphology (SKOV3), which is resistant to tumor necrosis factor. According to cytotoxicity tests, the rAFP3D-G2-DOX compound showed less toxicity to normal lymphocytes, but a significant decrease in cell viability, with an IC50 of 0.613 µM for SKOV3 cells. Furthermore, the combination demonstrated efficacy against human ovarian serous cystadenocarcinoma DOX-resistant SKVLB cells, indicating that receptor-mediated targeting and endocytosis may be used to overcome multidrug resistance [35].

However, despite their easy functionalization and distinctive benefits, dendritic polymers share a limiting characteristic with polymer therapeutics, because their multistep synthesis raises production costs [13].

### 4.4. Carbon-Based Nanomaterials

In addition to organic nanocarriers, inorganic materials with distinct physiochemical properties such as graphene oxide (GO), carbon nanotubes (CNTs), and carbon nanofibers are also promising candidates for generating nanoparticles [9,14,36]. These carbon-based materials can be post-chemically modified to become water-soluble nanocarriers. This has been found to increase their biocompatibility and decrease their toxicity because of their appropriate surface-to-volume ratio, thermal conductivity, and stiff structural characteristics [13,36].

#### 4.4.1. Nanographene Oxide (NGO)

The allotropic form of carbon, graphite, is arranged in a honeycomb lattice of hexagons. When graphite oxidizes, a monolayer of graphene oxide (GO) is created. Multilayered GO nanoparticles can be created by partially exfoliating this layer which has hydrophilic groups, including carboxyl, epoxy, and hydroxyl groups. The small size of the GO nanoparticles, their delocalized electrons, high surface area, near-infrared absorbance, and surface function are all thought to contribute to their advantageous qualities.

Because of their scalable synthesis, attractive physicochemical properties, ease of manufacture, and cost, NGOs have been used for fundamental research as well as pre-clinical and clinical applications in the field of nanobiotechnology. Depending on the equipment available and the experimental settings, several of the NGOs’ characteristics, including size, chemical composition, and surface charge density, have been modified for particular medical applications [11].

In one study, Qin et al. synthesized folic acid-conjugated graphene oxide with repeating units of poly (N-vinylpyrrolidone) (FA–NGO–PVP) for targeted chemo-photothermal therapy. Because of its strong π–π stacking and hydrophobic interactions, it achieved a high DOX loading efficiency, surpassing 100%. With 60% of the drug released over 70 h at an acidic pH (5.5), the DOX-loaded FA–NGO–PVP demonstrated pH-sensitive release, as opposed to 13% at a neutral pH (7.4). The amount of drug release increased with near-infrared (NIR) irradiation (808 nm, 2 W/cm^2^ for 3 min), reaching 70% in 10 h. At a DOX dose of 20 μg/mL, the combination of DOX-loaded FA–NGO–PVP with NIR irradiation produced a 90% cell-killing rate in Hela cells in vitro, which is much greater than the 70% attained by injecting DOX alone. This highlights the increased effectiveness of combined photothermal and chemotherapeutic strategies [37].

The Hela cell line was obtained from the tumor of a woman suffering from cervical cancer. Over the last 70+ years when these unique Hela cells have been grown in the laboratory, they have not only remained alive but have multiplied at an astonishing rate [38].

#### 4.4.2. Carbon Nanotubes (CNTs)

In addition to graphene oxide (GO), a promising class of inorganic carbon nanomaterials, is the family of carbon nanotubes (CNTs). They consist of graphene cylinders that have been coiled and have two end caps, similar to Buckminsterfullerene, which is a C_60_ fullerene with a truncated icosahedron structure made of twenty hexagons and twelve pentagons like a soccer ball [39]. CNTs have diameters ranging from 0.4 to 100 nm and lengths up to a few micrometers. Depending on the number of carbon layers they contain, CNTs can be divided into two primary categories: single-walled CNTs (SWCNT) and multi-walled CNTs (MWCNT). Every carbon cylinder or layer is made up of a graphene sheet [40]. Their distinct physicochemical characteristics, such as their hollow monolithic structure that can hold a large payload and has the potential to add any functional groups, and make them an appropriate, efficient and attractive delivery system for chemotherapeutic agents. By regulating the mode of drug release in vitro, it has been possible to reduce cytotoxicity while simultaneously increasing cellular uptake efficiency [13,14].

Since many medications and therapeutic compounds, such as anthracyclines, have aromatic rings in their structures, the hydrophobic surface of CNTs promoted π-π stacking interactions. Electrostatic (π-π stacking) interactions facilitate the anthracycline anticancer medication, DOX, to readily adsorb on the surface of CNTs [40]. For targeted DOX delivery, Lu et al. created folate-conjugated, magnetic multi-walled carbon nanotubes (FA-MN-MWCNTs). And when the DOX-to-MWCNT ratio was 2:1, the DOX loading efficiency on FA-MN-MWCNTs surpassed 96%, yielding a high drug loading content of 1.84 mg DOX/mg MWCNT. At pH 7.4 the regulated release of DOX was 14% over 192 h compared to the accelerated release of 71% at pH 5.3, confirming that DOX release was a pH-sensitive phenomenon. Furthermore, magnetic targeting produced targeted cytotoxic effects that effectively killed cells in the magnetic field’s limited area [41].

Cao et al. created a novel pH-responsive, polyethyleneimine-betaine (PEI-B) functionalized SWCNT system to improve the co-delivery and efficacy of survivin using small interfering RNA (siRNA) and DOX to further address drug resistance. Survivin is a member of the inhibitor of apoptosis protein family, which is highly expressed in most human cancers and has key roles in regulating cell division and inhibiting apoptosis by blocking caspase activation [42]. A high DOX loading capacity of 70.8% and an encapsulation efficiency of 94% was achieved using this technology. Because of tumor-specific acidity, the release of DOX was markedly accelerated under acidic conditions, with 30% released at pH 5.0 in 48 h as opposed to just 15% at neutral pH (7.4). Effective DOX penetration into the cytoplasm and nucleus of adenocarcinoma human alveolar basal epithelial (A549) cells was made possible by this pH-triggered strategy. The combination of chemotherapeutic and gene therapy components without significant harm to normal tissues was demonstrated by the DOX-SWCNT-siRNA complex (SPBB), which produced a tumor suppression rate in vivo of 69.22% (Figure 6), which was significantly greater than DOX alone at 48.31% [43].

### 4.5. Other Advanced Nanocarriers

The characteristics of mesoporous silica nanoparticles (MSNs), such as their uniform and adjustable pore size (50–300 nm), ease of surface modification, multiple functionalization combinations, high mechanical and thermal stability, and high biocompatibility and biodegradability, have drawn a lot of attention [44,45]. MSNs can hold large loads of drug molecules because of their “honeycomb” structure, which increases their surface area and pore volume [45]. PEG and chitosan-coated MSNs were investigated in one study as a DOX delivery vehicle. High DOX loading capabilities of 97.85% and 93.32% were demonstrated by the optimized formulations of 2% and 5% PEG-coated MSNs, respectively [46].

It was discovered that DOX release from MSNs was pH-dependent, with the maximum release taking place in acidic environments. According to in vitro drug release tests, electrostatic interactions were responsible for the approximately 76% of DOX that was released in an acetate buffer (pH 4) over the course of 72 h, compared to only 38% in a carbonate buffer (pH 10). The medication reached a plateau at over 80% release [47].

Smart coatings and targeting mechanisms have been included in later inventions to increase specificity and lessen off-target impacts. HeLa cells, for instance, showed a highly selective uptake of polydopamine-coated MSNs with tumor-specific ligands (FA and cyclic RGD peptide) that was 12 times greater than that of normal fibroblasts. With the clever MSN-DOX system 4000-fold more toxic to HeLa cells than to normal cells, this selective targeting resulted in greatly increased cytotoxicity against cancer cells [48]. When compared to free DOX, the MSN-DOX nanoconjugates showed enhanced bioavailability and therapeutic efficacy, making them a viable option for targeted cancer treatment.

When compared to free DOX, the MSN-DOX nanoconjugates showed enhanced bioavailability and therapeutic efficacy, making them a viable option for targeted cancer treatment.

Despite the fact that a significant number of nanocarriers are presently undergoing preclinical research, their clinical applicability is still limited. For instance, when NPs enter the body, it could be challenging to prevent cellular and autoimmune reactions in the blood of healthy tissues. In order to achieve immune evasion and prolong their biological half-life, recent developments in nanotechnology have focused on enhancing NP properties through surface modifications like PEG, as previously indicated. Even while these materials show encouraging potential, they can yet be improved in areas like targeting or immunological compatibility [49].

In order to support tumor-targeted therapy, a biomimetic camouflage technique using natural cell membranes (CMs) has gained more and more attention [49]. Cell membrane-coated NPs can “cloak” themselves by exploiting natural cell membranes, which delays immune system detection and extends circulation periods. This makes it possible for them to concentrate specifically in places like tumor sites where they might have the greatest impact. Additionally, the membranes that cover these NPs frequently preserve surface identifiers like proteins and antigens, which can enhance their capacity for targeting [50].

Hollow mesoporous organosilica nanoparticles coated with aptamer-functionalized red blood cells (Apt-RBC-HMOS@DOX) showed that the release of DOX was regulated and responsive to glutathione (GSH), with just 20% released at pH 7.4 and up to 50% at pH 5.4 over 48 h. This ensured stability in circulation and effective tumor targeting [51].

The cancer cell membrane-camouflaged gelatin-based nanogel (M-NG@DOX) showed improved tumor targeting and controlled DOX release in a distinct investigation. At physiological pH, the drug release was only 17.90% over 36 h, reducing premature leakage. With a 91.30% uptake efficiency, the biomimetic coating dramatically decreased macrophage uptake while promoting cellular internalization in 4T1 breast cancer cells. M-NG@DOX outperformed free DOX and uncoated nanogels in in vivo experiments, demonstrating greater tumor suppression with a tumor growth inhibition rate of 70.36%. These investigations and findings demonstrate the promise of bio-mimetic nanocarriers as a precise and efficient method of delivering chemotherapy drugs [52].

## 5. Limitations and Challenges

Without a question, nanocarriers have become a promising tool for targeted drug administration, with the potential to increase patient compliance and therapeutic success. However, there are many obstacles in the way of their development and clinical translation. Scaling up from laboratory to industrial scale has challenges in maintaining quality, homogeneity, and stability, which raises production costs. The manufacture of nanocarriers entails intricate and costly procedures. Furthermore, large-scale production is made more difficult and less consistent by the absence of defined manufacturing methods. Their practical application is further limited by instability caused by problems including drug-loading capacity decrease and nanoparticle aggregation. Clinically speaking, serious safety testing is required prior to human use due to worries about the biocompatibility and possible toxicity of specific nanocarriers [53,54,55]. Nanocarriers’ physicochemical characteristics affect their biocompatibility and toxicity in biological systems, therefore careful characterization is required to reduce any unintentional harm to healthy cells. Additionally, the development of clinically viable nanomedicines is made more difficult by differences between in vitro and in vivo drug release characteristics [3].

Since the FDA has not yet established precise standards for goods incorporating nanomaterials, regulatory approval continues to be a crucial barrier in the commercialization of nanomedicine. Because current regulatory frameworks are based on bulk material standards, the approval process is difficult and time-consuming. The transition of nanocarriers from research to market is further hampered by the technical difficulties of large-scale manufacturing [3]. Consequently, even though nanocarriers have a lot of potential, overcoming these restrictions is essential to maximizing their use in therapeutic settings.

## 6. Conclusions and Future Prospects

One of the most effective and popular chemotherapeutic drugs for treating a variety of tumors is still doxorubicin (DOX). In contrast to other traditional treatments like hormonal therapy and radiation, or more recent approaches like immunotherapy and targeted medicines, DOX has a proven, broad-spectrum cytotoxic effect with a distinct pharmacological profile. Numerous attempts have been undertaken to increase its safety and tolerance due to negative side effects that have limited its use. The development of innovative DOX-based nanocarriers, including as liposomes, micelles, dendrimers, and other delivery systems, for the treatment of cancer is explained and discussed in this article. Due to their unique size and structure, these nanocarriers shield DOX from burst release, offering excellent opportunities for controlled and sustained release as well as extending retention in tumor sites to optimize DOX’s energy efficiency. It is also important to remember that techniques like coating, ligand conjugation, and stimuli response enable these nanocarriers to target and reach cancer cells directly, minimizing negative effects and damage to healthy tissues. Recent research, such as biomimetic nanoparticles, focuses more on immunological and biocompatibility within the entire human system to further reduce risk.

As previously said, there are still many issues and obstacles that the nanocarriers must overcome. Several problems, like the biodegradability of inorganic nanocarriers and the complex process of polymeric nanocarriers call for further research and advancements in the field. The most important is to be approved and put on the market after completing all preclinical and clinical trials successfully. Furthermore, a better understanding of the clinical and pathophysiological differences that occur in patients is needed to determine the long-term effects of the FDA-approved, nanocarrier-based delivery systems.

## Figures and Tables

**Figure 1 bioengineering-12-00430-f001:**
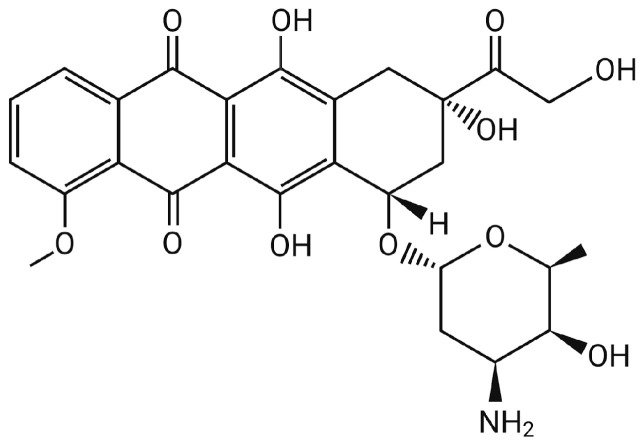
Molecular structure of doxorubicin.

**Figure 2 bioengineering-12-00430-f002:**
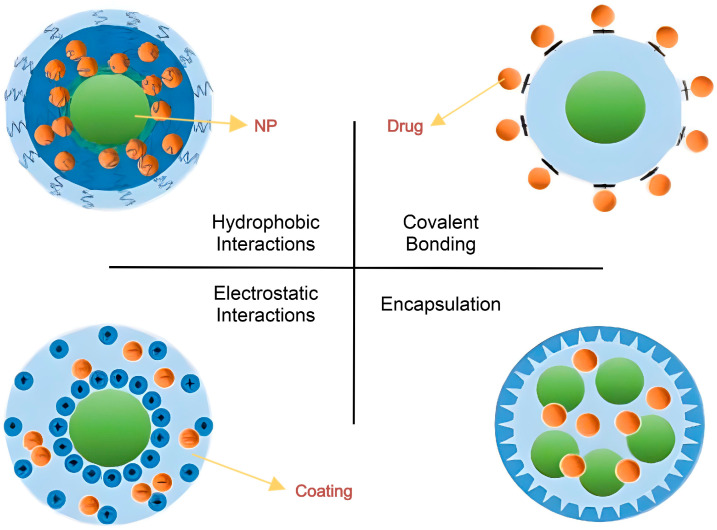
Methods of loading/bonding drugs into nanoparticles (NPs) (adapted from [13]).

**Figure 3 bioengineering-12-00430-f003:**
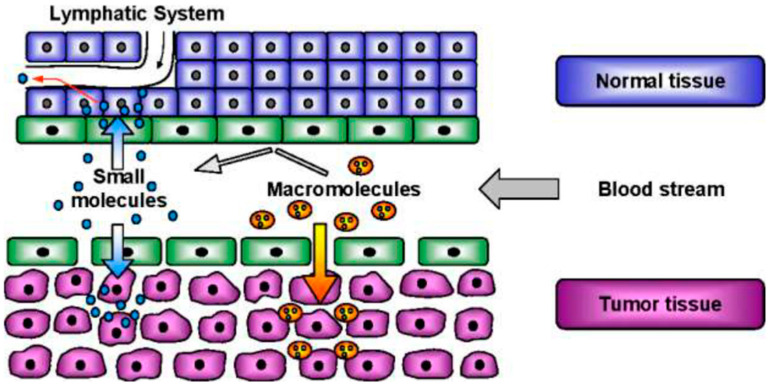
Nanoparticles’ passive targeting using enhanced penetration retention (EPR) (adopted from [1]).

**Figure 4 bioengineering-12-00430-f004:**
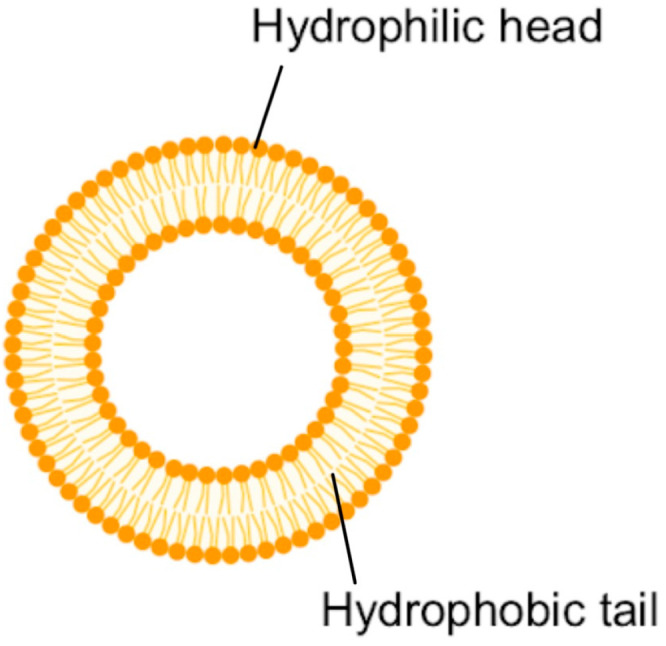
Structure of a typical liposome.

**Figure 5 bioengineering-12-00430-f005:**
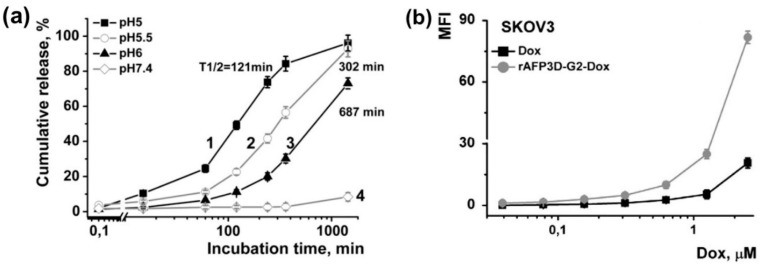
(**a**) DOX release from rAFP3D-G2-DOX conjugate at different pH values; (**b**) uptake of DOX and rAFP3D-G2-DOX by ovarian cancer (SKOV3) cells (Reprinted with permission from Ref. [35]. Copyright 2013 Elsevier).

**Figure 6 bioengineering-12-00430-f006:**
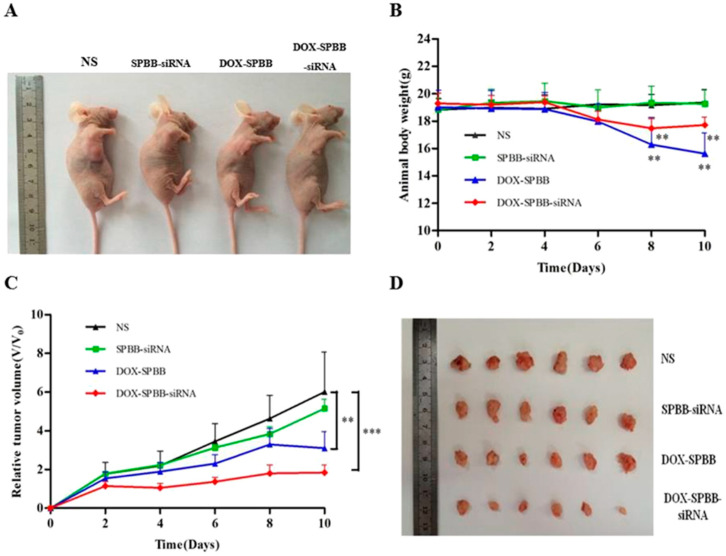
In vivo antitumor ability of SPBB loaded with DOX and/or siRNA in A549 tumor-bearing nude mice: (**A**) appearance of tumor growth; (**B**) changes in body weight over 10 days of treatment; (**C**) changes in relative tumor volumes over 10 days of treatment; (**C**) changes in relative tumor volumes over 10 days of treatment; (**D**) photograph showing relative size of tumor tissues after being treated for 10 days (** *p* < 0.01, *** *p* < 0.001 were considered statistically significant) (Reprinted with permission from Ref. [43]. Copyright 2019 American Chemical Society).

**Table 1 bioengineering-12-00430-t001:** Comparison of various nanocarrier systems.

Nanocarrier	Loading Efficiency	Advantages	Disadvantages
Liposomes	Moderate to high loading	High biocompatibility	Rapid reticuloendothelial system (RES) clearance
Micelles	High for hydrophobic drugs	Small size for deep tissue penetration	Potential premature drug release
Nanogels	Moderate loading	Excellent for stimuli-triggered release	Lower loading for very hydrophobic drugs
Dendrimers	Very high loading	High functionalization versatility	Complex synthesis and high cost
Nanographene Oxide (NGO)	Extremely high loading	Ease of surface modification for targeting	Potential toxicity and poor aqueous dispersibility
Carbon Nanotubes (CNTs)	High loading	Exceptional mechanical and thermal stability	Potential toxicity and poor water solubility
Mesoporous Silica Nanoparticles (MSNs)	Very high loading	High surface area and tunability	Possibility of long-term accumulation in organs
Cell Membrane-Coated Nanoparticles	Depending on the core nanoparticle	Enhanced targeting and reduced immunogenicity	More complex and costly to fabricate

**Table 2 bioengineering-12-00430-t002:** FDA-approved formulations and ongoing clinical studies involving DOX nanocarriers.

Formulation	Status	Indications	Description	References
Doxil^®^/Caelyx^®^ *	Approved (1995)	Ovarian cancer, AIDS-related Kaposi’s Sarcoma, Multiple Myeloma	The first FDA-approved nanodrug, pegylated liposomes, ammonium sulfate gradient	[20,21,22]
Myocet^®^	Approved (2000)	Metastatic breast cancer	Approved in Europe and Canada, non-pegylated liposomes, citrate transmembrane gradient	[23]
Lipodox^®^	Approved (2013)	Ovarian cancer, breast cancer, Kaposi’s sarcoma	Generic version of Doxil^®^, pegylated liposomes, ammonium sulfate gradient	[24,25]
ThermoDox^®^	Phase III	Hepatocellular carcinoma	Thermosensitive liposomes, pegylated liposomes, ammonium sulfate gradient	[18]
MM-302	Phase II	HER2-positive breast cancer	PEG-modified liposomes with doxorubicin and HER2-specific antibodies	[19,26]
2B3-101	Phase II	Breast cancer metastases in the brain	Liposomal doxorubicin hydrochloride with glutathione ligands	[19]
SP1049C	Phase III	Adenocarcinoma in the esophagus and gastroesophageal junction	A micellar formulation using Pluronics^®^ L61 and F127	[19,27]
NK911	Phase II	Metastatic pancreatic cancer	Another micellar formulation	[19,27]
FCE28068/PK1	Phase II	Breast, colorectal, and non-small cell lung cancers	A polymer conjugate, N-(2-hydroxypropyl)methacrylamide (HPMA) copolymer covalently bound to DOX by a peptidyl linker	[19,27]
FCE28068/PK2	Phase II	Liver cancer	Builds upon PK1 by adding galactosamine, asialoglycoprotein receptors	[19,27]

* This formulation is marketed as Doxil^®^ in the USA and Caelyx^®^ in the EU.

## Data Availability

Not applicable.

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
