# Peer review of "Advances in Targeted Delivery of Doxorubicin for Cancer Chemotherapy"

_bioengineering, 2025, doi:10.3390/bioengineering12040430_

Round 1
Reviewer 1 Report
Comments and Suggestions for Authors
The authors have done a great job of investigating the strategies for DOX release for the cancer treatment. However, this manuscript needs to improve in some aspects, and comments are given in the attached document.

Author Response
For review article
Response to Reviewer 1's Comments
- Summary
Thank you very much for taking the time to review this manuscript. Please find the detailed responses below and the corresponding revisions in the re-submitted files.
- Response to Comments and Suggestions for Authors
Comments 3&6: While the review effectively covers the findings, it neglects to explore the limitations and problems of these nanocarriers, particularly with practical factors such as cost, scalability, and clinical applicability.
Response 3&6: Thank you for pointing this out. We agree with this comment. Therefore, we added an independent section for this part, which is on page 14, section 5.
Comment 4: Recent developments in nanocarriers including inorganic silica especially MSNs for the release of DOX have been extensive, including those that would add more attention to the manuscript.
Response 4: Agree. We have, accordingly, added relevant MSN research to enrich recent developments in nanocarriers. The introduction and related experiments of MSNs are on page 13, section 4.5.
Comment 5: Supplement a comparative table summarizing the advantages and disadvantages of the various nanocarrier systems discussed. This would make the information easier to understand and provide a better reference for readers.
Response 5: Thank you for pointing this out. We agree with this comment. Therefore, we added a comparative table in front of the introduction to all the nanoparticles in detail, which is on page 5, section 4.
Comment 7: Elaborate the conclusion section focusing on the DOX effectiveness for the cancer treatment over other strategies.
Response 7: Agree. We have, accordingly, revised the conclusion part to emphasize this point.
Reviewer 2 Report
Comments and Suggestions for Authors
The review article titled “Advances in Targeted Delivery of Doxorubicin for Cancer Chemotherapy” addresses a highly significant and timely topic in cancer therapeutics. While the manuscript is engaging and holds great potential, several critical gaps and areas for improvement need to be addressed before it can be considered for acceptance. The following points outline the key suggestions to enhance the quality and impact of the article:
Major Observations:
1. Figure Improvements:
o Clearly indicate the sources for all images (e.g., “Adopted from___”) and improve their resolution/quality.
o Include a graphical summary illustrating the key concepts of DOX delivery via nanocarriers for better visual appeal.
o Enhance figure legends for clarity, particularly for Figures 4, 6, 8, and 13.
o Correct text alignment in Figure 4 to ensure readability.
2. Content and Accuracy:
o Verify the details of the nitrogen drying step in the ammonium sulfate method (e.g., Drying of Lipid under nitrogen).
o The statement “Despite a significant number of clinical studies, no DOX nanocarriers have been approved by the FDA” is incorrect. Provide accurate data and include a table summarizing FDA-approved formulations and ongoing clinical studies for DOX nanocarriers.
3. Organizational Issues:
o Separate the discussion of in vitro and in vivo studies for each nanocarrier.
o Create a comparative table summarizing key outcomes from in vitro and in vivo studies based on previous literature.
o Maintain consistent depth of explanation across different nanocarriers. Some, like liposomes, are extensively discussed, while others are covered minimally.
4. Title Consistency:
o Revise and differentiate Titles 3 (Nanocarriers) and 4 (Nanocarriers Used for Targeted DOX Delivery) for logical coherence.
5. HeLa Cell Paragraph:
o The paragraph detailing the history of HeLa cells in the Nanographene oxide (NGO) section, while informative, detracts from the focus. Condense this to focus on its relevance to NGO applications.
6. Abstract:
o Rewrite the abstract to ensure better alignment with the manuscript’s key findings and provide a clearer summary of its contributions.
Minor Observations:
1. Grammar and Syntax:
o Proofread the manuscript to correct grammatical errors and awkward phrasing (e.g., “DOX delivery causes cellular damage in both healthy and cancerous cells”).
2. References:
o Update the references by incorporating more recent studies, especially those published post-2020, to reflect advancements in DOX nanocarriers.
3. Explanation of Studies:
o For Figure 10, clarify that the graph represents an in vitro drug release study.
4. Comparative Analysis:
o Add a table comparing different nanocarriers (e.g., liposomes, micelles, dendrimers) based on parameters like loading efficiency, release profiles, and therapeutic efficacy.
Author Response
- Summary
Thank you very much for taking the time to review this manuscript. Please find the detailed responses below and the corresponding revisions in the re-submitted files.
- Response to Comments and Suggestions for Authors
Comment 1:
Figure Improvements:
o Clearly indicate the sources for all images (e.g., “Adopted from___”) and improve their resolution/quality.
o Include a graphical summary illustrating the key concepts of DOX delivery via nanocarriers for better visual appeal.
o Enhance figure legends for clarity, particularly for Figures 4, 6, 8, and 13.
o Correct text alignment in Figure 4 to ensure readability.
Response 1: Thank you for pointing these out. We agree with these comments. Therefore, we deleted some unnecessary figures and modified the original Figure 4, which is now Figure 2 on page 4.
Comment 2: The statement “Despite a significant number of clinical studies, no DOX nanocarriers have been approved by the FDA” is incorrect. Provide accurate data and include a table summarizing FDA-approved formulations and ongoing clinical studies for DOX nanocarriers.
Response 2: Agree. We have, accordingly, added the related table to show developments and the current situation of nanocarriers, which is on page 6.
Comment 3:
Separate the discussion of in vitro and in vivo studies for each nanocarrier.
o Create a comparative table summarizing key outcomes from in vitro and in vivo studies based on previous literature.
Response 3: Thank you for your suggestion. However, I am sorry that I don't know how to discuss them separately. I am not sure if the comparative table should be for each nanoparticle or the big summary one.
Comment 4: Revise and differentiate Titles 3 (Nanocarriers) and 4 (Nanocarriers Used for Targeted DOX Delivery) for logical coherence.
Response 4: Agree. We have, accordingly, revised the title to differentiate.
Comment 5: The paragraph detailing the history of HeLa cells in the Nanographene oxide (NGO) section, while informative, detracts from the focus. Condense this to focus on its relevance to NGO applications.
Response 5: Agree. We have, accordingly, revised the HeLa cells part, which is on page 11, section 4.4.1, paragraph 4.
Comment 6: Rewrite the abstract to ensure better alignment with the manuscript’s key findings and provide a clearer summary of its contributions.
Response 6: Agree. We have, accordingly, revised the abstract part.
Reviewer 3 Report
Comments and Suggestions for Authors
Dear Authors,
The problem of DOX cardiotoxicity is quite well known for a long period. In its current state, the Manuscript cannot be considered for publication in the Journal. Nevertheless, I believe it can be resubmitted after its thorough revision and expansion to a standard Review article size of at least 100-300 references being analyzed.
Also I have some additional comments on the Manuscript in case of its resubmission:
1. The Manuscript contains some statements without any reference cited, that is normal for a research article, but not for a review. For example:
"Due to its excellent efficacy at low doses,...";
"DOX delivery causes cellular damage in both healthy and cancerous cells, which leads to myelosuppression, nausea, and vomiting shortly after administration", etc.
2. The goal of the Review should contain its novelty in comparison to the previously published papers including reviews on the same subject.
3. The Manuscript contains no original (not taken from the literature sources!) summarizing and comparative tables and schemes. These elements should be added to stress the Review originality and to make it useful for the readers of the Journal.
4. The Section 4 does not contain the latest results shown with a use of biomimetic nanostructures (cell-coumuflaged nanocarriers, nanocarriers with a protein shell, etc.), therefore this issue has to be considered in a revised version of the Manuscript.
5. The quality (resolution) of some figures is poor and needs to be enhanced.
6. The overall quantity of figures (19) is excess for such a compact review article, therefore I recommend to omit some not necessary figures.
7. The Conclusions and Future Prospects section needs to be extended considering the new analytical results shown in the Manuscript.
8. The overall references amount (31) is not enough for a comprehensive review article. Also there are not enough newer papers and books on the subject (after 2023) in the References list.
9. Some refs (e.g. [1], [8] and [9]) are cited too many times in the Manuscript, including the figures from these previously published articles, therefore its novelty and originality are not obvious in its current state.
Author Response
- Summary
Thank you very much for taking the time to review this manuscript. Please find the detailed responses below and the corresponding revisions in the re-submitted files.
- Response to Comments and Suggestions for Authors
Comments 1:
The Manuscript contains some statements without any reference cited, that is normal for a research article, but not for a review. For example:
"Due to its excellent efficacy at low doses,...";
"DOX delivery causes cellular damage in both healthy and cancerous cells, which leads to myelosuppression, nausea, and vomiting shortly after administration", etc.
Response 1: Agree. We have, accordingly, revised the sentence and added the reference on page 2, line 5.
Comment 3: The Manuscript contains no original (not taken from the literature sources!) summarizing and comparative tables and schemes. These elements should be added to stress the Review originality and to make it useful for the readers of the Journal.
Response 3: Agree. We have, accordingly, added relevant comparative tables to distinct different nanocarriers, which is on page 5.
Comment 4: The Section 4 does not contain the latest results shown with a use of biomimetic nanostructures (cell-coumuflaged nanocarriers, nanocarriers with a protein shell, etc.), therefore this issue has to be considered in a revised version of the Manuscript.
Response 4: Thank you for pointing this out. We agree with this comment. Therefore, we added the introduction and research of the biomimetic cell membrane-coated nanoparticle, which is on page 14, paragraph 2.
Comment 7: The Conclusions and Future Prospects section needs to be extended considering the new analytical results shown in the Manuscript.
Response 7: Agree. We have, accordingly, revised the conclusion part to show the contributions and key findings better.
Comment 8: The overall references amount (31) is not enough for a comprehensive review article. Also there are not enough newer papers and books on the subject (after 2023) in the References list.
Response 8: Agree. We have, accordingly, added more references to enrich the content. However, we intended to write one small review article in the first place, and it is a little difficult to achieve 100-300 references.
Comment 9: Some refs (e.g. [1], [8] and [9]) are cited too many times in the Manuscript, including the figures from these previously published articles, therefore its novelty and originality are not obvious in its current state.
Response 9: Agree. We have, accordingly, deleted and replaced some text and figures that were cited from references cited too many times.
Round 2
Reviewer 2 Report
Comments and Suggestions for Authors
The author has revised the manuscript for most of the comments and suggestions. However, authors need to revise and respond to the following minor revisions.
- Grammar and Syntax:
- Proofread the manuscript to correct grammatical errors and awkward phrasing (e.g., “DOX delivery causes cellular damage in both healthy and cancerous cells”).
- References:
- Update the references by incorporating more recent studies, especially those published post-2020, to reflect advancements in DOX nanocarriers.
- Explanation of Studies:
- For Figure 10, clarify that the graph represents an in vitro drug release study.
- Comparative Analysis:
- Add a table comparing different nanocarriers (e.g., liposomes, micelles, dendrimers) based on parameters like loading efficiency, release profiles, and therapeutic efficacy.
Author Response
Thank you for your support and reminders. I am sorry that I have made these revisions but did not mention them last time.
Comment 1: Proofread the manuscript to correct grammatical errors and awkward phrasing (e.g., “DOX delivery causes cellular damage in both healthy and cancerous cells”).
Response 1: Agree. We have revised them accordingly by changing the expression, which can be found on page 2 in red font.
Comment 2: Update the references by incorporating more recent studies, especially those published post-2020, to reflect advancements in DOX nanocarriers.
Response 2: Thank you for your suggestion. We accepted and added more references, and most of them were published post-2020, which can be proven in the reference section with yellow highlighting.
Comment 3: For Figure 10, clarify that the graph represents an in vitro drug release study.
Response 3: Agree. We have accordingly deleted related content to avoid confusion.
Comment 4: Add a table comparing different nanocarriers (e.g., liposomes, micelles, dendrimers) based on parameters like loading efficiency, release profiles, and therapeutic efficacy.
Response 4: Agree. We have added the loading efficiency part in the table on page 5 with yellow highlighting. However, it is difficult for us to summarize release profiles and therapeutic efficacy because it depends on different particles and techniques.
Reviewer 3 Report
Comments and Suggestions for Authors
Dear Authors,
Thank you for revising the Manuscript. In my opinion, now it is quite suitable for publication in its current state. My only recommendation is to take into account copyright issues, since the figure captions do not contain any license statements (e.g. permission from the publishers or CC-BY license, or something). Nevertheless, this task is out if the Reviewer's expertise and has to be solved with the Editorial Office.
Author Response
Thank you so much for your time and approval. The figure copyright issue is also my big concern. I am unsure if I should contact the original authors to ask for permission or if the editor may act on this. Thank you again for your support and for raising this problem.